# Contingent Social Interaction Does Not Prevent Habituation towards Playback of Pro-Social 50-kHz Calls: Behavioral Responses and Brain Activation Patterns

**DOI:** 10.3390/brainsci12111474

**Published:** 2022-10-31

**Authors:** Annuska Berz, Camila Pasquini de Souza, Markus Wöhr, Sebastian Steinmüller, Maria Bruntsch, Martin K.-H. Schäfer, Rainer K. W. Schwarting

**Affiliations:** 1Behavioral Neuroscience, Experimental and Biological Psychology, Faculty of Psychology, Philipps-University Marburg, 35032 Marburg, Germany; 2Center for Mind, Brain and Behavior, Philipps-University Marburg, 35032 Marburg, Germany; 3Department of Pharmacology, Biological Sciences Building, Federal University of Parana, Curitiba 81530-000, PR, Brazil; 4KU Leuven, Faculty of Psychology and Educational Sciences, Research Unit Brain and Cognition, Laboratory of Biological Psychology, Social and Affective Neuroscience Research Group, B-3000 Leuven, Belgium; 5Leuven Brain Institute, KU Leuven, 3000 Leuven, Belgium; 6Institute of Anatomy and Cell Biology, Faculty of Medicine, Philipps-University Marburg, 35032 Marburg, Germany

**Keywords:** c-fos expression, GABA, nucleus accumbens, anterior cingulate cortex, Wistar rat, social approach, ultrasonic vocalization

## Abstract

Rats, which are highly social animals, are known to communicate using ultrasonic vocalizations (USV) in different frequency ranges. Calls around 50 kHz are related to positive affective states and promote social interactions. Our previous work has shown that the playback of natural 50-kHz USV leads to a strong social approach response toward the sound source, which is related to activation in the nucleus accumbens. In male Wistar rats, the behavioral response habituates, that is, becomes weaker or is even absent, when such playback is repeated several days later, an outcome found to be memory-dependent. Here, we asked whether such habituation is due to the lack of a contingent social consequence after playback in the initial test and whether activation of the nucleus accumbens, as measured by c-fos immunohistochemistry, can still be observed in a retest. To this end, groups of young male Wistar rats underwent an initial 50-kHz USV playback test, immediately after which they were either (1) kept temporarily alone, (2) exposed to a same-sex juvenile, or (3) to their own housing group. One week later, they underwent a retest with playback; this time not followed by social consequences but by brain removal for c-fos immunohistochemistry. Consistent with previous reports, behavioral changes evoked by the initial exposure to 50-kHz USV playback included a strong approach response. In the retest, no such response was found, irrespective of whether rats had experienced a contingent social consequence after the initial test or not. At the neural level, no substantial c-fos activation was found in the nucleus accumbens, but unexpected strong activation was detected in the anterior cingulate cortex, with some of it in GABAergic cells. The c-fos patterns did not differ between groups but cell numbers were individually correlated with behavior, i.e., rats that still approached in response to playback in the retest showed more activation. Together, these data do not provide substantial evidence that the lack of a contingent social consequence after 50-kHz USV playback accounts for approach habituation in the retest. Additionally, there is apparently no substantial activation of the nucleus accumbens in the retest, whereas the exploratory findings in the anterior cingulate cortex indicate that this brain area might be involved when individual rats still approach 50-kHz USV playback.

## 1. Introduction

Rats, which are very social animals, use several sensory routes of communication, including vocalizations, especially in the ultrasonic range. These ultrasonic vocalizations (USV) are known to serve as situation-dependent socio-affective signals (for reviews, see [1,2]. Specifically, juvenile and adult rats emit USV in two major call classes. Calls of one of them, termed 22-kHz USV, are typical for aversive situations such as predator exposure. They probably express a negative affective state and serve as alarm signals to others (e.g., [3,4]). The other, so-called 50-kHz calls, are emitted in appetitive situations, for example, rough-and-tumble play or mating (e.g., [5,6]). They are thought to reflect the sender’s positive affective state [7] and can serve to initiate, maintain and coordinate social interactions among conspecifics (e.g., [8,9,10,11,12,13,14,15]).

Several years ago, we introduced a playback paradigm to measure the approach-eliciting properties of 50-kHz USV [16]. Their effectiveness to investigate approach in response to playback of 50-kHz USV, but not noise or 22-kHz calls, has repeatedly been demonstrated in male and female Wistar and Sprague Dawley rats, supporting the hypothesis that they serve as social contact calls (e.g., [16,17,18,19,20,21,22,23,24]); for reviews, see [25,26]. Additionally, the effectiveness of 50-kHz USV playback is clearly dependent on the animals’ developmental stage, because the approach was found to be more pronounced in juveniles than adults [16], that is, the age of rough-and-tumble play [27]. Given the effectiveness to study social approach in a recipient, the playback paradigm has repeatedly been applied to investigate the outcomes of environmental manipulations, i.e., social isolation and enrichment [20,21], as well as possible social deficits in animal models of human neuropsychiatric and neurological disorders [23,28,29,30,31,32,33].

With respect to the neuronal mechanisms of 50-kHz USV playback effects, we could show that they are related to dopamine (DA) and opioid function in the brain. Regarding opioids, we found that the systemic administration of the agonist morphine promoted, whereas the antagonist naloxone reduced, playback-induced approach [17]. Similar effects were obtained in case of drugs with strong DAergic actions, such as amphetamine, which increases catecholamine availability, particularly DA concentrations in the synaptic cleft [34,35], which were enhanced, whereas the D2 receptor antagonist haloperidol impaired such an approach ([24], but see [32]). These psychopharmacological findings were supported by neurochemical ones showing social approach induced by 50-kHz USV to be related to increased DA release in the nucleus accumbens (NAcc; [36], a brain area well-known for its critical role in motivated behavior (e.g., [37])). Finally, studies were performed to analyze local immediate early gene activity in response to 50-kHz USV playback. Compared to the effects of presumably danger-signalling 22-kHz calls, 50-kHz USV playback predominantly led to decreased neuronal activity (for example, in the central amygdala) and moderate activation, which occurred in the frontal part of the secondary motor cortex M2 [38]; termed FrA according to an older edition of the atlas of Paxinos and Watson [39] as well as in the NAcc [29]. Given the prominent role of the NAcc both in the emission of 50-kHz USV and in the approach behavior induced by such stimulus, it was assumed that the NAcc serves a critical role in the approach-eliciting properties of 50-kHz USV.

Importantly, approach to 50-kHz USV playback was found to be most prominent during its initial exposure compared to a retest performed several days later [12,24,40]. Interestingly, this effect was observed in male Wistar but not Sprague Dawley rats [24], indicating that it is strain-dependent. In Wistar rats, repeating the same kind of playback led to reduced approach or even no further response in a retest. This intriguing habituation effect was found to be memory-dependent, since scopolamine, an amnesia-producing antagonist of muscarinic acetylcholine receptors, prevented habituation when given immediately after the initial test, that is, during the presumptive memory consolidation phase [18]. Furthermore, we found that repeated 50-kHz USV playback loses its effectiveness to increase extracellular DA levels in the NAcc [36], but that habituation of approach can be prevented by the administration of amphetamine prior to the retest [24].

Apart from these findings, the pronounced habituation effect of Wistar rats remained largely unexplained. Considering the prominent role of 50-kHz as pro-social signals, one could assume that the habituation effect reflects some kind of extinction; that is, the rats no longer approach the signal source during the retest, since they have learned from the initial test that the social signal is not followed by a social consequence.

The present experiment was designed to test this hypothesis, namely, that rats would still show an approach to 50-kHz playback in a retest if they experienced a social encounter immediately after the initial test. In order to do so, three groups of rats received an identical playback procedure with 50-kHz USV. In two groups, this playback was immediately followed by a social encounter, either with an unfamiliar same-sex juvenile, or the test subject’s housing group, whereas another group temporarily remained alone after playback. The playback procedure was repeated in the same way one week later, now not followed by any social consequence but perfusion for later c-fos immunohistochemistry. In short, our expectation that social interaction after the initial test would prevent habituation in the retest was not supported, since none of the three groups showed an approach in the retest that was comparable to that of the initial test. The c-fos analysis was somehow in line with this outcome, since there was also no significant activation in the NAcc. Unexpectedly, dense c-fos labeling was observed in the anterior cingulate cortex (ACC), part of which was located in GABAergic cells. This labeling did not differ on the group level, but individual and group-independent analyses yielded a rather strong positive correlation with residual approach in the retest.

## 2. Subjects and Methods

### 2.1. Animals and Housing

We used thirty-six juvenile male Wistar rats obtained from Charles-River, Germany, which weighed 146.1 ± 2.7 g (which equals about 6 weeks of age) at the start of the experiment. Thirty rats served as the experimental subjects and the others served as social partners after the initial playback test. The rats were maintained under standard housing conditions in groups of 5–6 rats per cage (polycarbonate, macrolon type IV, size 380 × 200 × 590 mm with high steel covers), with water and food available ad libitum, a 12/12 light-dark cycle with lights on at 7 am, and humidity ranging between 32 and 50%. Prior to the start of the experiment, rats had seven days of acclimatization, followed by a standard protocol of handling on three consecutive days (five minutes each). All procedures had been approved by the ethical committee of the local government (Regierungspräsidium Gießen, Germany, TVA Nr. 35–2018).

### 2.2. Experimental Design

Thirty animals were randomly assigned to three groups (n = 10 each), which underwent a repeated playback procedure consisting of two tests (termed initial test and retest) with a one-week interval in between (see Figure 1 for illustration). Each playback test lasted 20 min: 15 min of no acoustic stimulation, followed by 5 min of 50-kHz USV playback. Immediately after the initial test, each rat was transported to a separate room, where it was either kept for 10 min in a small housing cage with fresh bedding (termed Empty Cage), a small housing cage containing a non-familiar male juvenile rat (termed Social Partner), or the group cage from which the tested rat originated (termed Group Cage). Thereafter, the rats from conditions Empty Cage and Social Partner were returned to their respective group cages. After the retest, each rat was kept singly for 60 min in a small housing cage with fresh bedding, followed by the brain removal procedure.

### 2.3. Acoustic Stimuli and Experimental Setups

For playback, we used series of 50-kHz USV, which had been recorded from an adult male Wistar rat (ca. 350 g) during exploration of a cage containing scents from a recently separated cage mate (for details see [12]. This stimulus material was composed of a sequence lasting 3.5 s, presented in a loop. Each sequence contained 13 50-kHz calls (total calling time: 0.90 s), with 10 of them being frequency-modulated and 3 flat (for details, see [16]. Peak amplitude was about 70 dB (measured from a distance of 40 cm), which is within the typical range of 50-kHz USV [23,28]. These USV were presented through an ultrasonic loudspeaker (ScanSpeak, Avisoft Bioacoustics), which had a frequency range of 1–120 kHz with flat frequency response (+/−12 dB) between 15 and 80 kHz. Sounds were played via an external sound card with a sampling rate of 192 kHz (Fire Wire Audio Capture FA-101, Edirol, London, UK) and a portable ultrasonic power amplifier with a frequency range of 1–125 kHz (Avisoft Bioacoustics, Glienicke/Nordbahn, Germany).

### 2.4. Radial Maze Playback Paradigm

Social approach induced by 50-kHz USV was assessed on a radial eight-arm maze (arms 40.5 × 9.8 cm, see Figure 2), elevated 52 cm above the floor, as described by [16]. The maze was monitored by a Basler aca camera placed 150 cm centrally above this. The ultrasonic speaker was placed 20 cm away from the end of one arm and an additional, but inactive speaker was arranged symmetrically at the opposite arm as a visual control. To start a test, the given rat was placed into the center of the maze, facing away from both ultrasonic speakers. Testing took place between 7 and 17 h under red light (~10 lux), with no other rats present in the testing room. After each test, the equipment was cleaned thoroughly with acetic acid 0.1% and dried.

### 2.5. Behavioral Analysis

Behavior was recorded via the video camera and analyzed using EthoVison XT (Version 13, Noldus, Wageningen, The Netherlands). Locomotor activity was measured in terms of distance traveled (in cm). Additionally, the numbers of entries into the three arms proximal and the three arms distal to the active ultrasonic loudspeaker and the times spent thereon were separately quantified. As in previous studies, proximal measures served for stimulus-directed activity, i.e., approach to 50-kHz USV playback (e.g., [24,41]).

### 2.6. Immunohistochemical Analysis

Animals were deeply anesthetized with pentobarbital and transcardially perfused with 0.9% saline and 4% paraformaldehyde in 0.1M phosphate buffer, pH 7.4. Brains were removed, postfixed and cryo-protected in 30% sucrose/0.01M phosphate-buffered saline (PBS) and frozen on dry ice. Coronal sections of 40 µm were sliced on a cryostat and subsequently processed for immunocytochemistry. For permeabilization, the sections were washed in 0.01 M PBS, followed by a 30 min incubation in 0.3% H_2_O_2_ (80,702, Carl Roth, Karlsruhe, Germany) to block endogenous peroxidase activity and then incubated for 10 min in 0.2% Triton ×100 (3051.3, Carl Roth). Non-specific binding sites were blocked with 5% normal goat serum (NGS-Vector S-1000) in 50 mM PBS for 30 min. Sections were incubated with a mouse monoclonal antibody to c-fos (sc-2712432 von Santa Cruz Biotechnology Inc., Heidelberg, Germany) at a final dilution of 1:1000 (1%NGS) for 48–72 h at 4 °C. After washing, sections were incubated with biotinylated secondary antibody (Sigma B0529 goat-anti-mouse, Schnelldorf, Germany) diluted1:1000 in PBS-T for 90min, washed and incubated for 90 min with avidin–biotin–peroxidase complex (Vectastain Elite ABC kit; PK-6100, Vector Laboratories, Burlingame, CA, USA). Immunoreactions were then visualized by incubation in 0.125 mg/mL DAB (3,3′-diaminobenzidine tetrahydrochloride hydrate, D5637, Sigma-Aldrich, Schnelldorf, Germany).

c-Fos expression was qualitatively screened using microscopic photos taken by a BX 63 Olympus microscope. c-Fos-positive cells were then quantified using the software ImageJ according to histologically defined criteria of the Paxinos and Watson rat atlas [42]. Counting was done in a 3.27 mm^2^ oval shape area for the NAcc and a stipulated 0.42 mm^2^ triangle area for the anterior cingulate cortex. Prior to this, images were adjusted by turning them into 8-bit versions and removing the low-intensity information of the image via background subtraction and thresholding. Then, c-fos positive cells were automatically counted using the plugin Cell Count (ImageJ) inside the determined region of interest (ROI).

#### Double Immunofluorescence

To check whether c-fos labeled cells had a GABAergic phenotype, additional coronal brain sections were subjected to double immunofluorescence for c-fos and glutamic acid decarboxylase (GAD). For that purpose, four animals, which exhibited the highest numbers of c-fos labeled cells, were selected and prepared as above. Primary antibodies to c-fos (1:100; 1% NGS) and GAD (rabbit anti-GAD 65 + 67, 1:200, 1%NGS) were co-applied overnight at 16 °C. After extensive washing, immunoreactions for GAD were visualized with goat anti-rabbit antibodies conjugated to Alexa488 (1:400, Dianova, Hamburg, Germany). c-Fos immunoreactions were visualized by a two-step procedure using a biotinylated donkey anti-mouse followed by streptavidin-Cy3 conjugate (1:400, Thermofisher, Dreieich, Germany).

Immunofluorescence results were documented with the Olympus fluorescence microscope B × 63, using Cy3 channel for c-fos-positive cells and FITC channel for GAD-positive cells. The ACC was identified by histologically defined criteria of the Paxinos & Watson rat atlas [42]. For visualization, the software ImageJ was used to colour the photos in red for c-fos and green for GAD. By merging the pseudo-coloured images, co-labeled cells could be identified. Before analysis, background noise was reduced by equally adjusting the brightness and contrast of the used photos. Finally, positive cells were manually counted, distinguishing between c-fos-positive cells, GAD-positive cells and cells co-positive for c-fos and GAD.

### 2.7. Statistical Analysis

Response to playback: Behavior during the 5 min either before or during playback was analyzed either as absolute values or as change scores (according to [18]), i.e., the responses to playback were calculated by subtracting entry or time measures proximal or distal to the acoustic source during the 5 min before stimulus presentation from those during the 5 min of stimulus presentation. To test for stimulus effects, these scores were analyzed with one-sample *t*-tests (versus 0). Paired *t*-tests were used for comparing proximal versus distal changes in arm entries or times spent thereon. For comparing experimental groups, ANOVAs for repeated measures with the factor treatment (3 levels: Empty Cage, Social Partner, Group Cage) and the dependent factor location (proximal versus distal) were calculated. Locomotor activity during the first 15 min of the retest were analyzed with an ANOVA for repeated measures using the factor group (3 levels: Empty Cage, Social Partner, Group Cage) and the dependent factor minutes (15 levels). c-Fos results were analyzed using ANOVA for repeated measures with the factor treatment (3 levels: Empty Cage, Social Partner, Group Cage) and the dependent factor hemisphere (left, right). To correlate individual behavior with c-fos labeling, we used Pearson’s correlation coefficient. Data are presented as means ± SEM (standard error of the mean).

## 3. Results

### 3.1. Behavior—Initial Test

Prominent approach behavior was seen during the initial exposure to playback of 50-kHz USV. Since behavior of the latter experimental groups did not differ during this playback test (all *p*-values > 0.05), their results were pooled and presented together both, in terms of absolute 5-min values either before or during playback (Figure 3) and as change scores (during minus before; Figure 4). Comparing proximal versus distal arm entries or times spent in these arms by means of *t*-tests (2-tailed) showed that proximal versus distal values did not differ during the 5 min before playback (*p*-values > 0.05). During 50-kHz USV playback, the rats responded with the expected approach, that is, they showed more arm entries into proximal than distal arm entries (T_28_ = 9.272, *p* < 0.001) and spent more time on proximal versus distal arms (T_28_ = 10.440, *p* < 0.001).

Analysis of the change scores (two-tailed *t*-tests versus 0) supported these results since proximal arm entries (*p* = 0.002), as well as proximal times (*p* < 0.001), increased, and distal arm entries (*p* < 0.001) and times spent (*p* < 0.001) decreased during the 5 min of playback as compared to the preceding 5 min.

In addition, locomotor activity during the 15 min prior to and the 5 min during 50-kHz playback was analyzed in terms of distance travelled (Figure 5). This analysis showed that locomotor activity declined over time (factor minutes: F_1,28_ = 866.484, *p* < 0.001). This decline was observed throughout the test except for a transient increase during the 1st playback minute (post hoc paired two-tailed *t*-test between min 15 and 16, T_28_ = 3.130, *p* = 0.005).

Behavior in the subsequent phase, i.e., exposure to the housing group, an unfamiliar social partner or an empty cage, was not quantified in detail, since the three different conditions are too different for relevant comparisons. Nevertheless, the video and ultrasonic recordings were checked by a trained observer, especially with respect to the condition Social Partner, since the respective experimental animals were exposed to an unfamiliar conspecific, which could have led to unwanted aggressive encounters. Similar to the Group Cage condition, this was never the case. Typically, the two animals were rather active during this exposure phase in terms of social (such as facial contacts, nape contacts, ano-genital sniffing or following) and non-social behaviors (such as locomotion, rearing or grooming). Additionally, 22-kHz calls, which might reflect aversion, were never observed, except for two calls in one Group Cage test. On the other hand, 50-kHz calls were rather frequent and, interestingly, especially in the Social Partner condition, which again indicates that this social encounter was rather appetitive for the experimental animal.

### 3.2. Behavior—Retest

During the 5 min prior to 50-kHz USV playback, the number of proximal versus distal arm entries (Figure 6, left) did not differ (factor location: F_1_ = 0.643, *p* = 0.430). Additionally, there was no effect of treatment (F_2,27_ = 1.842, *p* = 0.178) and no interaction between the factors location and treatment (F_2,27_ = 2.600, *p* = 0.093). During subsequent 50-kHz USV playback (Figure 6, right), there was a general difference between proximal versus distal arm entries (F_1_ = 10.354, *p* = 0.003), i.e., more proximal than distal entries, but no difference between groups (F_2,27_ = 0.407, *p* = 0.670) nor was there an interaction between location and groups (F_2,27_ = 1.080, *p* = 0.354). The change scores of arm entries (Figure 7) did not yield significant changes in proximal entries during the 5 min of playback as compared to the preceding 5 min (*t*-tests, two-tailed, all *p*-values > 0.05), but decreases in distal entries in the groups Empty Cage (T_9_ = −3.108, *p* = 0.013) and Group Cage (T_9_ = −2.862, *p* = 0.019) and a trend for such an effect in the group Social Partner (T_9_ = −2.094, *p* = 0.066).

For the measure of times spent, there was no general difference between proximal and distal arms prior to 50-kHz USV playback (Figure 8 left; factor location: F_1,27_ = 0.989, *p* = 0.329) and no difference between treatments (F_2,27_ = 0.233, *p* = 0.793), but an interaction between location and treatments (F_2,27_ = 4.939, *p* =0.015). Subsequent two-tailed *t*-tests showed more proximal than distal time in group Group Cage (T_9_ = 3.029, *p* = 0.042; *p*-value corrected for multiple comparisons), but no differences in the other two groups. During subsequent 50-kHz USV playback (Figure 8, right), there were no significant effects (factor location: F_1,27_ = 0.355, *p* = 0.556, factor treatment: F_2,27_ = 0.529, *p* = 0.595, interaction: F_2,27_ = 0.524, *p* = 0.598). The changes in scores of times spent (Figure 9) did not yield any significant differences in the three groups regarding times spent during as compared to before 50-kHz USV playback (all *p*-values > 0.05).

In addition, locomotor activity during the 15 min prior to and the 5 min of 50-kHz USV playback was analyzed (Figure 10). Similar to the initial test, locomotor activity declined over time (factor minutes: F_19, 38_ = 45.714, *p* < 0.001) and there was a trend of a general difference between treatments (F_2,27_ = 3.187, *p* = 0.057), but no interaction between minutes and treatments (F_38,513_ = 0.798, *p* = 0.802). Unlike the initial test, there was no evidence for an increase in locomotion during the first minute of playback.

### 3.3. Immunohistochemistry

NAcc: The numbers of c-fos labeled cells in the NAcc (Figure 11b) were rather low and varied considerably between subjects. These numbers did not differ between treatments (F_2_ = 0.495, *p* = 0.615), left and right hemispheres (F_1_ = 0.579, *p* = 0.453), nor was there an interaction between treatments and hemispheres (F_2,27_ = 0.654, *p* = 0.528). Additionally, there was no evidence for differences in labeling between shell and core of the NAcc (data not shown). Furthermore, we asked whether there were correlations between individual cell numbers and approach measures in the retest preceding brain removal, but the respective correlation coefficients were low in all groups (between −0.212 and −0.079) and far from significance levels (*p*-values between 0.261 and 0.678).

Further exploratory analyses: When examining the brain sections with respect to c-fos labeling in the NAcc, a striking density of c-fos positive cells was observed in the dorsal neocortex, namely in the ACC and sometimes also the adjacent M2 (Figure 12a). Therefore, we decided to also examine these patterns, which led to the following results: the numbers of labeled cells per area in ACC were clearly higher than those in the NAcc. Similar to the NAcc, we tested the ACC data using an ANOVA for repeated measure, but here as a descriptive measure of effect, which should not be interpreted in terms of statistical significance since the ACC outcomes had not been expected *a priori*. Again, however, there were no indications for treatments or hemispheric differences, nor interactions between these factors (all *p*-values > 0.100).

We also correlated individual cell numbers and approach behavior in the retest, which led to an interesting outcome (see Figure 12d): the behavioral change score (proximal arm time in response to 50-kHz calls compared to the preceding 5 min of no playback) was positively correlated with individual cell counts in the left ACC (r_28_ = 0.603, *p* = 0.001); that is, the more time individual animals spent in the arms proximal to 50-kHz USV playback, the higher the number in the left ACC. No such strong correlations were observed in case of the right ACC (r_29_ = −0.081, *p* = 0.676), nor in case of the arm entries’ change scores (left ACC: r = 0.315, *p* = 0.103, right ACC: r = −0.181, *p* = 0.347).

Finally, and to further anatomically characterize c-fos labeling in the ACC, we additionally applied GAD immunostaining to visualize GABAergic cells, for which we selected four rats with the highest numbers of c-fos-positive cells. By means of immunofluorescence microscopy, we locally counted the numbers of c-fos-positive cells, GAD-positive cells and cells co-positive for both, c-fos and GAD (see Figure 12c for an example). Since there were no apparent differences between hemispheres in these examples, we pooled them, which led to the following results: (A) The numbers of c-fos-positive cells were higher than those of GAD-positive cells (50.73 ± 8.94 versus 21.75 ± 3.49, n = 8), and (B) cells positive for c-fos as well as GAD accounted for 12.24 ± 4.46% of all c-fos positive cells.

## 4. Discussion

The present experiment was designed to test the hypothesis that the previously reported habituation of social approach of male Wistar rats to repeated playback of 50-kHz USV [18,24] may be due to the lack of a social consequence after the initial playback experience; that is, we asked whether rats would still show approach to 50-kHz playback in a retest if they had experienced a social encounter immediately after the initial test. Additionally, we wanted to test whether possible approach after repeated playback might be related with increased activation in the NAcc as labeled by c-fos immunostaining. The results provide only some evidence for the social hypothesis and none for the NAcc. Unexpectedly, however, we found strong c-fos activation in the ACC, which was positively individually correlated with approach after repeated playback. These results will be discussed in the following.

### 4.1. Behavioral Findings

The initial playback test yielded the expected results, namely, pronounced approach to 50-kHz USV playback in terms of times spent and arm entries, that is, the animals spent more time proximal than distal to the loudspeaker during playback and showed more entries into these proximal arms. These effects were obvious in the change scores, namely, increases in proximal entries and times spent during as compared to before playback together with decreases in distal activities. The locomotor data also showed a transient increase in response to playback. These results are largely in line with our previous ones [16,17,18,19,20,21,24,34,35].

During the retest, the measure of arm entries yielded a general effect during playback, i.e., more proximal than distal entries, and this effect was largely driven by the groups Empty Cage and Group Cage. Taken as such, one could assume that playback in the retest again led to an approach response in these groups, but the change scores showed that these outcomes were probably due to distal decreases rather than proximal increases, in contrast to proximal increases together with distal decreases in the initial test. The distal decreases during the retest may have reflected habituation of radial maze exploration, which, in case of proximal arms, might have been prevented by 50-kHz USV playback. The locomotor patterns over the 15 min prior to playback are in line with such an assumption, since activity declined over time, probably reflecting habituation to maze exposure. Additionally, and in contrast to the initial test, there was no evidence for a transient increase in locomotion in response to playback. On the other hand, one should consider that the significant changes in distal scores were found in groups Empty Cage and Group Cage, which obviously does not support our social hypothesis, since effects in the retest should be observed in the experimental conditions Group Cage and Social Partner, where two kinds of social consequences had been provided after the first playback but not in the condition Empty Cage, which lacked contingent social consequences.

Additionally, we found that the measure of times spent led to a result pattern, which differed from that of arm entries in the retest. Prior to playback in the retest, there was already an effect in the group Group Cage, which showed more time in the proximal than the distal arms. No such effect was observed in the other two groups. During subsequent playback, there were no significant effects in terms of absolute values and there were also no significant effects in the change scores when comparing behavior during versus prior to playback. Therefore, one can conclude that the measure of time spent did not provide any evidence of 50-kHz USV-induced approach in the retest (in contrast to the initial test), which seems to be in line with our previous studies [18,24]. Unlike previous work, however, one experimental group (Group Cage) spent more time in the proximal arms prior to playback in the retest. This result, if not chance, cannot be a response to playback, since it preceded it, but may reflect its expectation. The fact that no such patterns were found in our previous studies or in the two other groups tested here may be due to the social consequence after initial 50-kHz USV playback in the experimental condition Group Cage, i.e., where the experimental rats were exposed to the familiar housing group after playback in the first test, which may have served as a reinforcing sequel of the probably socially incentive 50-kHz USV playback. Such social consequences were not contingently provided in our previous studies and in the present group Empty Cage, and were perhaps insufficient in the present group Social Partner, where the experimental rat had contingent social contact but with only one unfamiliar conspecific. The inspection of videos and ultrasonic recordings, however, do not support the latter assumption, since the two animals in the condition Social Partner intensively interacted with each other and showed no evidence of aggressive encounters. Additionally, 22-kHz were not emitted at all, whereas 50-kHz calls were very frequent, which also indicates that these encounters were appetitive; that is, they served the experimental purpose. In summary, the approach measures during playback did not yield evidence in favour of our social hypothesis, whereas there was some indication for this in one social group prior to playback in the retest. This latter outcome was unexpected and should be further examined in the future. Finally, one should also consider that the social interactions in the groups Social Partner and Group Cage were provided after 5 min of playback and required that the test animals were manually removed from the playback apparatus. This removal might have acted as a stressor, which, in turn, could have impaired the possible appetitive effects of 50-kHz USV playback. Additionally, it is possible that extinction of the playback effects already occurred during the 5 min of playback, so that the later social consequences could not reinforce these. One should consider that the situation in case of the condition Social Partner differed from that of Group Cage in the sense that the animals were cage mates of the experimental rat in case of Group Cage, whereas the animal in the Social Partner conditions was an unfamiliar one. Therefore, the two experimental conditions did not only differ in terms of the number of social subjects with which to interact but also in terms of novelty. Such novelty probably played a larger role in case of Social Partner; additionally, one can assume that the experimental rat somehow entered the territory of the other one, which might be anxiogenic. Nevertheless, our qualitative inspections showed that the two rats in the condition Social Partner substantially interacted with each other with no evidence of aggressive encounters, no aversive 22-kHz calls but very frequent 50-kHz calls. This indicates that this condition was clearly appetitive for our experimental rats; that is, it served our purpose of an appetitive encounter after the initial 50-kHz playback.

Since our social feedback hypothesis could not substantially be supported by the present experiment, one has to consider alternative explanations, none of which have been tested. One possible explanation for the habituation effect to 50-kHz playback in the retest could be that habituation was specific to the acoustic stimuli, since these were identical in both tests, or that some kind of spatial learning occurred since the position of the active speaker was kept constant in both tests.

### 4.2. Anatomical Findings

In the NAcc, we found no evidence for selective activation patterns, since cell numbers did not differ between groups, nor were there correlations with individual approach patterns in the retest. This brain area was selected for anatomical analysis, since activations in response to 50-kHz USV playback had been shown in case of extracellular DA activity [36] as well as c-fos labeling [29]. Our own previous c-fos study, however, had only shown a trend for such an effect, which was probably due to the small sample sizes [38]. Importantly, the positive playback results were obtained in case of an initial 50-kHz USV playback experience and may no longer be observed after repeated experiences, as tested here. Evidence in favour of this assumption was already found in our voltammetry study [36], since enhanced DA release vanished with repeated exposures of 50-kHz USV playback. This result rules out the idea that initial activations reflected basic sensory processing in the NAcc, since these should still be effective in a retest. Alternatively, one could argue that the initial activation patterns may be due to novelty, which surely plays a role in case of the NAcc, but novelty (or its lack with repeated playback) cannot be the sole mechanism, since the novel playback of 22-kHz calls did not lead to increased DA release [36] or increased c-fos labeling in the NAcc [38]. These results also argue against stimulus salience in general: similar to 50-kHz calls, 22-kHz calls are also clearly salient, since they serve, among others, as danger signals (e.g., [3,4]). In contrast, appetitive salience and the link to approach action may be important, i.e., functions, for which the NAcc is known (for example, in terms of “from motivation to action” [37]).

In contrast to the NAcc, a striking but unexpected pattern of c-fos labeling was found in the ACC, an area with somehow inconsistent nomenclature in the rat: Thus, the terminology has changed over editions of the Paxinos and Watson rat brain atlas, as reviewed in [42a]. What had been named Cg 1 in the fourth edition was adapted to then human brain, and therefore re-labeled as 24b (also termed areas 24 b and b’, i.e., 24 b more anterior and 24 b’ more posterior, according to [43]. The dense c-fos labeling in our present study seems to be largely located in 24 b (formerly termed Cg 1) but may also comprise medial parts of the adjacent secondary motor cortex M2. As a first attempt to characterize the neurochemical phenotype of these cells, we applied GAD co-labeling to visualize GABAergic cells and found relatively few cells that were positive for c-fos and GABA, which is somehow surprising, given that GABAergic neurons constitute a rather substantial portion in the neocortex (e.g., [44]).

Before discussing the possible reasons for the c-fos labeling, one should emphasize that this finding is of exploratory nature, since we had no *a priori* hypothesis for the ACC. Therefore, these results should not be evaluated in terms of statistical significance and the following assessments should be handled with care. Thus, the activation patterns in the ACC were clearly denser in terms of cells/area as compared to the NAcc, but also did not differ between treatment groups. In contrast to the NAcc, however, a pronounced positive correlation between individual cell numbers and the change scores (proximal arm times) in the retest was found, which was apparently not driven by a specific experimental group; that is, a certain social experience after the first playback was apparently not critical for this outcome. Alternatively, individual aspects might have played a role: Kabbaj and Akil [45] reported that rats classified as high responders based on their locomotor activity in a novel environment showed a higher expression of c-fos mRNA in the ACC than low-responder rats after exposure to a light–dark anxiety test. These rats also showed more time in the bright section of the environment, indicating low anxiety and/or high novelty-seeking. Possibly, the correlation between approach to playback and ACC c-fos labeling observed in our study might reflect a similar mechanism; that is, some of our rats were perhaps more curious and/or less anxious than others, which might have enhanced the likelihood to re-approach the source of 50-kHz USV playback in the retest.

Interestingly, in our prior study [38], where the histological analysis was performed after the initial test, c-fos labeling in the ACC (there termed Cg 1) after 50-kHz USV playback was also analyzed and reported to be descriptively higher than in rats with no or 22-kHz USV playback. The fact that those differences were not significant might have been due to the rather small sample sizes. Therefore, it remains unclear whether the present ACC effect is specific to the retest or may also occur during an initial test. Additionally, the reason why correlations between c-fos activation and approach were found in the left, but not right, ACC, is unclear. This could reflect some kind of functional lateralization in this prefrontal area, and such prefrontal lateralization is apparently not uncommon in both rodents [46] and humans (for reviews see [47,48] and plays a role in emotions, stress as well as motor functions [49,50,51,52].

Functionally, van Heukelum et al. [43] reviewed several studies, which implicated area 24 b/Cg1 in attention, decision-making (see also [53]) and reward, but also pain and negative affect. Other authors emphasized the role of the ACC in social functions (for review see [54]. As a prerequisite for that, the ACC receives substantial sensory inputs from several modalities. Additionally, the ACC can relay such signals and, therefore, can modulate their impact on other brain structures, as shown in case of auditory processing in mice [55]. Such sensory signals are used for decision-making and consequent action; for example, that “motor actions coordinated by the ACC are selected based on expected rewards” (cited from Burglos-Robles et al. 2019). Apps et al. [56], for example, concluded that this brain area contributes “to social cognition by estimating how motivated other individuals are”. This hypothesis seems to fit with to our results, since rat 50-kHz calls probably signal a positive affective and pro-social state, that is, one that helps to initiate or maintain conspecific social interaction. The ACC is, in fact, linked to USV in the rat in case of both, 50-kHz USV emission and processing of perceived USV [57,58,59]. This somehow parallels the well-known situation in humans and monkeys, where parts of the ACC can be considered vocalization areas (for review see [60]). Saito & Okanoya [59] analyzed event-related potentials based on local field potentials in the ACC of awake rats and reported that ultrasonic signals elicited large signal amplitudes, especially when the sounds were in the 50-kHz range and frequency-modulated. They suggested that the ACC can process the emotional content of USV. Perhaps, our present ACC results reflect such subject-dependent processing and the link to appropriate action proposed here. This hypothesis is speculative and requires further attention in future studies.

Limitations: As pointed out before, the present findings in case of the ACC are exploratory; that is, they were not expected and the experiment was, therefore, not specifically designed for them. Thus, our study did not include specific ACC control groups, for example, one without playback in the retest, to further characterize the mechanisms that led to the intense c-fos labeling in this brain area.

## 5. Conclusions

Our data do not provide support for the hypothesis that approach habituation, as gauged in a playback retest several days after the initial playback of natural 50-kHz USV is due to the lack of social contact after playback, but indicate that the ACC seems to be involved when individual rats still approach 50-kHz USV in such a retest. Further studies are required to identify the factors that determine habituation and its individual variability after playback of 50-kHz.

## Figures and Tables

**Figure 1 brainsci-12-01474-f001:**
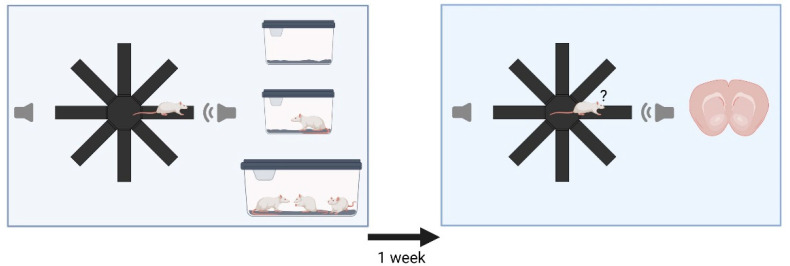
Graphical illustration of the experimental design. The animals underwent an initial playback test (**left box**), after which they were exposed either to an empty cage, a non-familiar social partner, or their respective group. Playback in the retest (**right box**) was identical to that of the initial test but was followed by the procedure of brain removal.

**Figure 2 brainsci-12-01474-f002:**
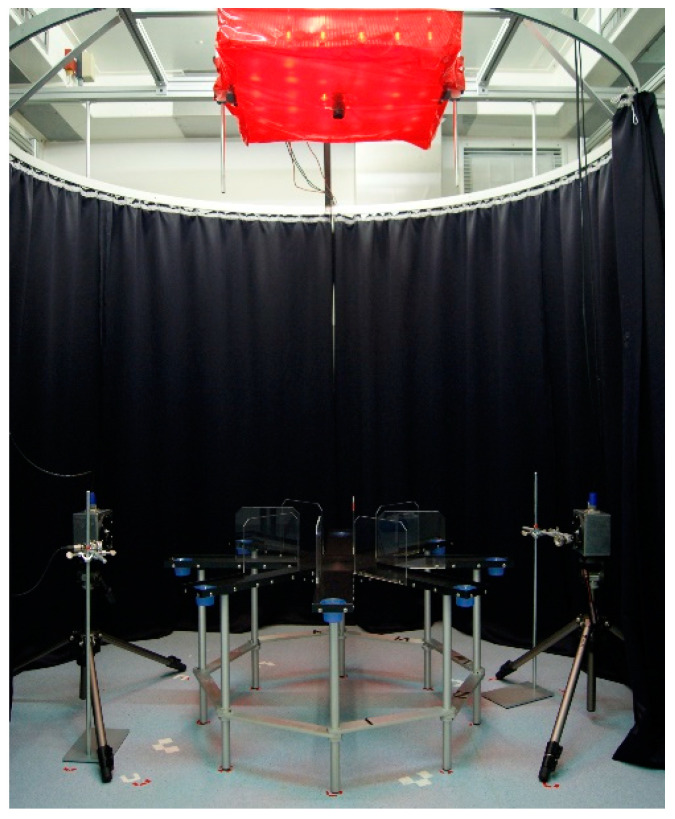
Photograph of the radial eight-arm maze used for playback. For details see text. Please note that the photo was taken under conditions of bright light for better visibility of technical details, whereas the tests were always conducted under red light conditions.

**Figure 3 brainsci-12-01474-f003:**
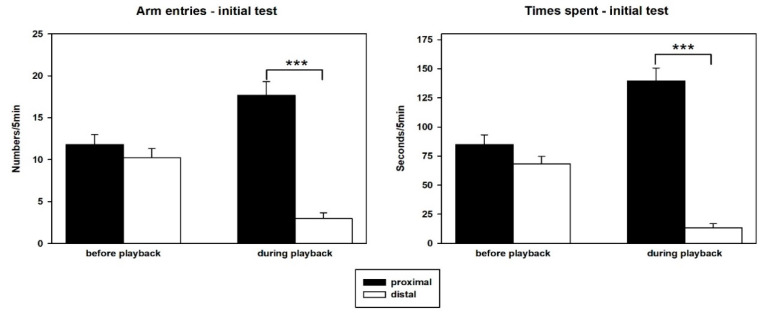
Behavioral responses elicited by playback of 50-kHz USV during the initial test. Behavioral responses were quantified as arm entries (**left graph**) and times spent on arms (**right graph**) proximal to (black) or distal from the sound source (white) during the 5 min of 50-kHz USV playback. Data are presented as means + SEM. *** *p* < 0.001 for proximal versus distal values.

**Figure 4 brainsci-12-01474-f004:**
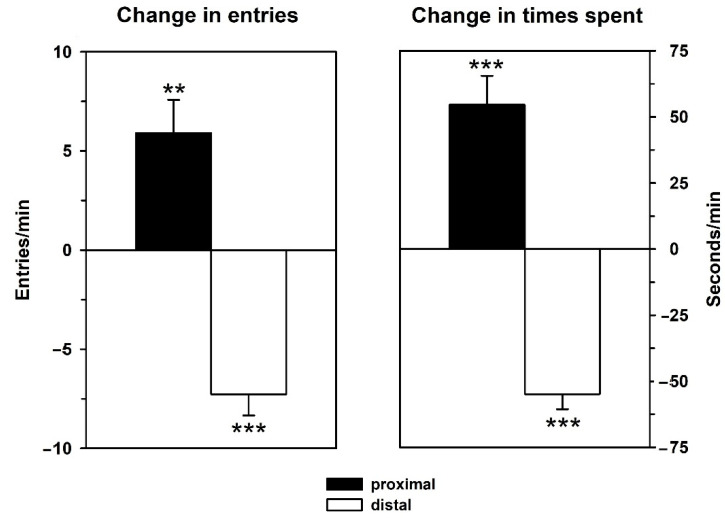
Behavior in the initial test expressed as change scores. Change scores for arm entries (**left graph**) or time spent on arms (**right graph**) proximal to (black bars) or distal from the sound source (white bars) were calculated by subtracting entry and time measures during the 5 min before stimulus presentation from those during the 5 min of stimulus presentation. Data are presented as means ± SEM. ** *p* < 0.01, *** *p* < 0.001, compared to baseline.

**Figure 5 brainsci-12-01474-f005:**
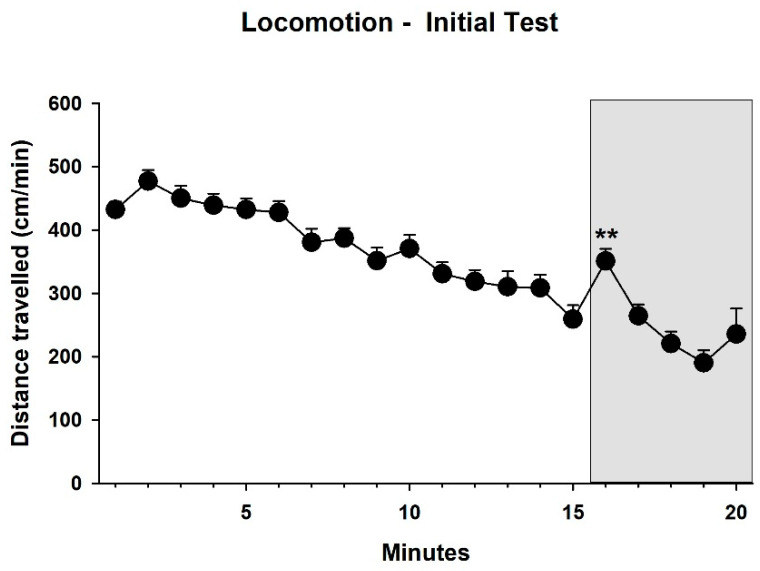
Locomotor activity in cm/min in the initial test during the 15 min before and the 5 min (indicated by grey background) during 50-kHz playback. During the 1st min of playback, locomotor activity increased as compared to the prior minute (** *p* = 0.005). Data are presented as means + SEM.

**Figure 6 brainsci-12-01474-f006:**
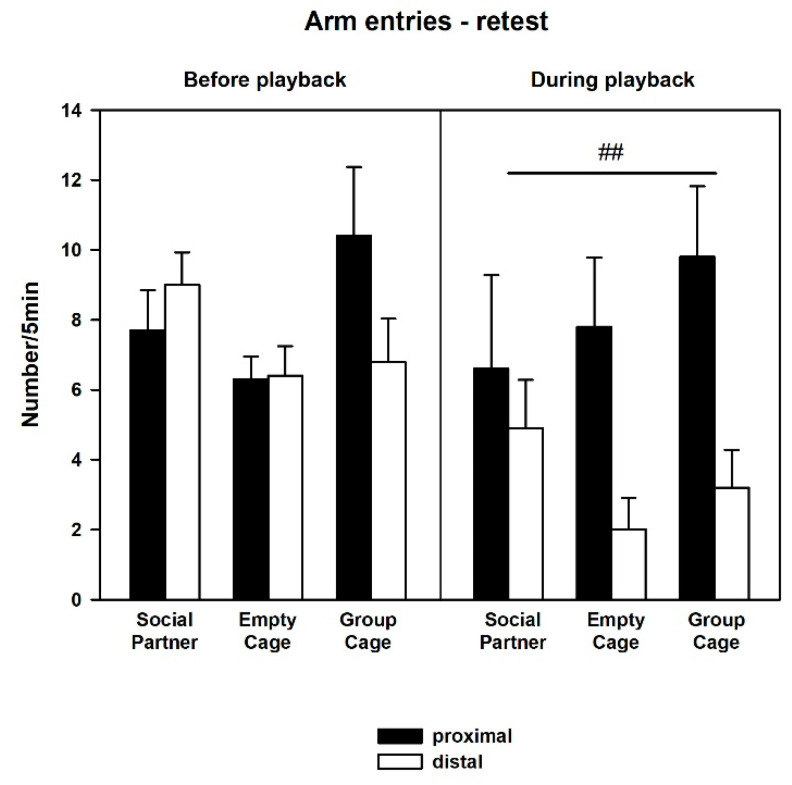
Proximal (black) or distal (white) arm entries in the retest before (**left**) or during (**right**) playback of 50-kHz USV in animals, which had access to a conspecific (Social Partner), no social contact (Empty Cage) or their cage mates (Group Cage) immediately after playback in the initial test. Data are presented as means + SEM. ## indicates a general difference (*p* = 0.003) between proximal and distal values.

**Figure 7 brainsci-12-01474-f007:**
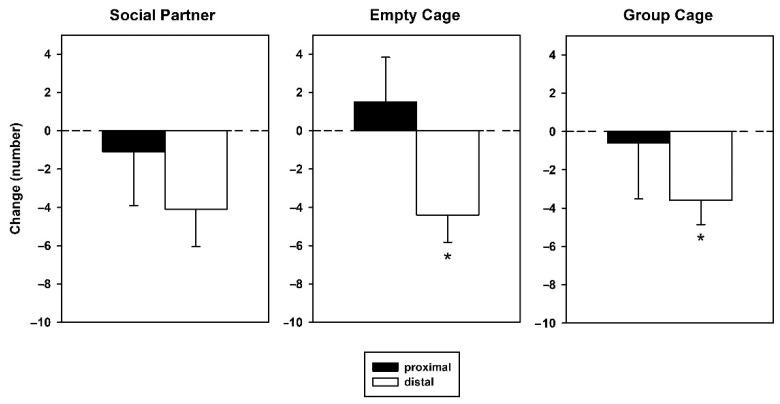
Arm entries in the retest expressed as change scores in arms proximal to (black bars) or distal from the sound source (white bars) in animals, which had access to a conspecific (Social Partner), no social contact (Empty Cage) or their cage mates (Group Cage) immediately after playback in the initial test. Change scores were calculated for arm entries proximal (black bars) or distal from the sound source (white bars) by subtracting entries during the 5 min before stimulus presentation from those during the 5 min of stimulus presentation. Data are presented as means ± SEM. * *p* < 0.05, compared to baseline.

**Figure 8 brainsci-12-01474-f008:**
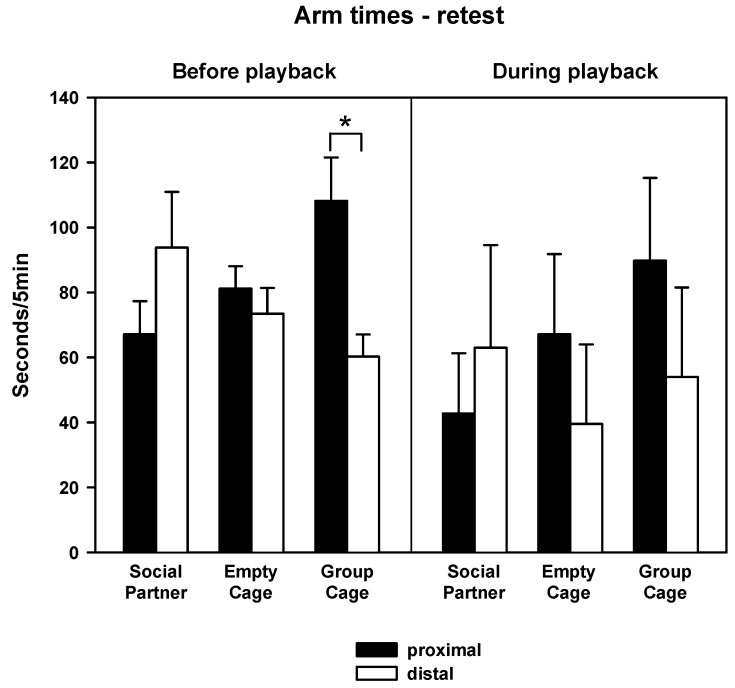
Proximal (black) or distal (white) arm times in the retest before (**left**) or during (**right**) playback of 50-kHz USV in animals, which had access to a conspecific (Social Partner), no social contact (Empty Cage) or their cage mates (Group Cage) immediately after playback in the initial test. Data are presented as means + SEM. * *p* < 0.05 for proximal versus distal values.

**Figure 9 brainsci-12-01474-f009:**
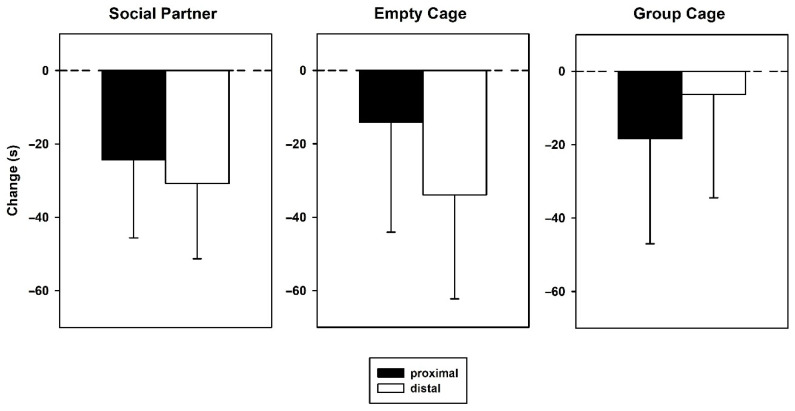
Arm times in the retest expressed as change scores in arms proximal to (black bars) or distal from the sound source (white bars) in animals which had access to a conspecific (Social Partner), no social contact (Empty Cage) or their cage mates (Group Cage) immediately after playback in the initial test. Change scores were calculated for times in arms proximal (black bars) or distal from the sound source (white bars) by subtracting entry and time measures during the 5 min before stimulus presentation from those during the 5 min of stimulus presentation. Data are presented as means + SEM.

**Figure 10 brainsci-12-01474-f010:**
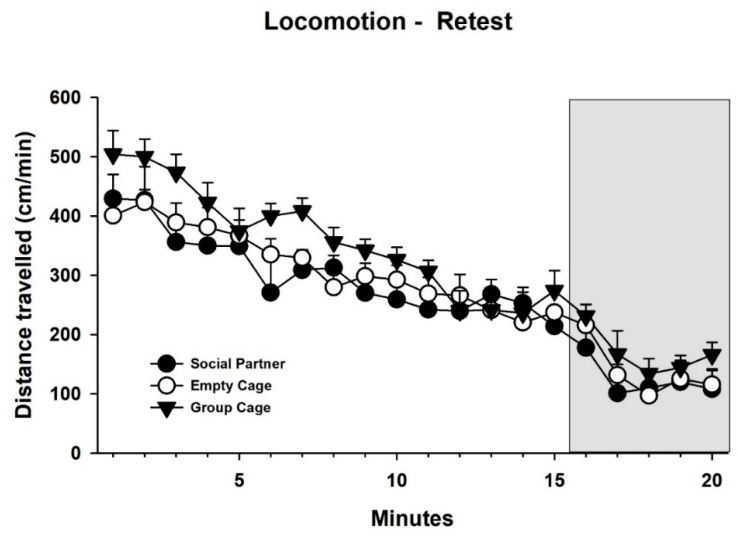
Locomotor activity in cm/min in the retest during the 15 min before and the 5 min during (indicated by grey background) playback in animals, which had access to a conspecific (Social Partner, black circles), no social contact (Empty Cage, white circles) or their cage mates (Group Cage, black triangles) immediately after playback in the initial test. Data are presented as means + SEM.

**Figure 11 brainsci-12-01474-f011:**
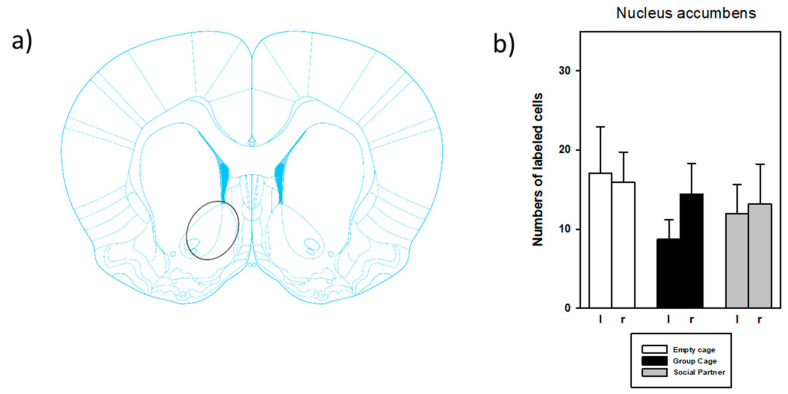
(**a**) Schematic section according to plate 16 (Bregma 2.04 mm) of the atlas of Paxinos and Watson [42] showing the region of interest for counting c-fos-positive cells in the NAcc. (**b**) Numbers of c-fos-positive cells in the NAcc of the left (l) and right hemisphere (r) in animals, which had access to a conspecific (Social Partner, grey bars), no social contact (Empty Cage, white bars) or their cage mates (Group Cage, black bars) immediately after playback in the initial test. Data are presented as means + SEM.

**Figure 12 brainsci-12-01474-f012:**
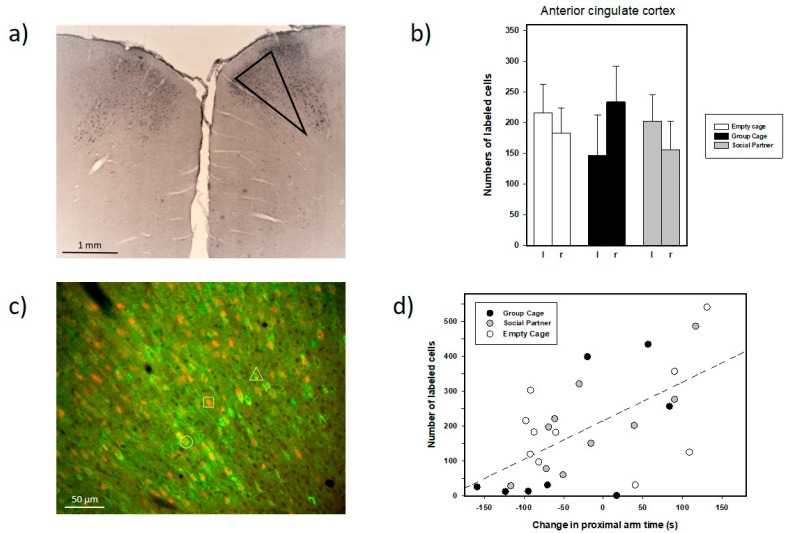
(**a**): Histological example (approximately plate 16, i.e., Bregma 0.84 mm, of the atlas of Paxinos and Watson [42] showing dense c-fos labeling in the ACC and the adjacent M2. (**b**) Numbers of c-fos-positive cells in the ACC of the left (l) and right hemisphere (r) in animals, which had access to a conspecific (Social Partner, gray bars), no social contact (Empty Cage, white bars) or their cage mates (Group Cage, black bars) immediately after playback in the initial test. Data are presented as means + SEM. (**c**) c-Fos and GAD immunolabeled cells in the ACC. Microscopic image depicting double-immunofluorescent labeling taken within the triangulated area shown in a. c-Fos immunoreactive neurons are labeled in red (Cy3) and GAD neurons in green (Alexa488): Note single labeled neurons positive for c-fos (indicated by square) or for GAD (indicated by triangle) and cells co-positive for c-fos and GAD (indicated by circle) located in the ACC. (**d**) Scatter plot plus regression line depicting the individual relationships between the number of c-fos-positive cells in the ACC and approach, i.e., the change in proximal arm times during playback of 50-kHz USV in the retest in animals, which had access to a conspecific (Social Partner, gray circles), no social contact (Empty Cage, white circles) or their cage mates (Group Cage, black circles) immediately after playback in the initial test.

## Data Availability

The data presented in this study are available on request from the corresponding author.

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
