# Peer review of "Contingent Social Interaction Does Not Prevent Habituation towards Playback of Pro-Social 50-kHz Calls: Behavioral Responses and Brain Activation Patterns"

_brainsci, 2022, doi:10.3390/brainsci12111474_

Round 1

Reviewer 1 Report

This is a fantastic well written paper that tests if approach habituation to playback is a result of not having social stimuli after the test. The experiment is well done, and interesting, it provides new insights into the habituation effect of playback. I have minor comments mostly about discussion points but overall the manuscript is a welcome and novel addition to the field.

Abstract

Line 20 – 50 kHz should have a hyphen to be continuous

Introduction

Line 59 – why does it say but see before the last reference

Line 70 – The Knutson reference is for anticipation of play and USVs, the correct reference for the juvenile period would be (D.H. Thor, W.R. Holloway Jr. Sex and social play in juvenile rats (Rattus novegicus) J. Comp. Psychol., 98 (1984), pp. 276-284)

Line 114 – please clarify the hypothesis discussed

Methods

Line 130 – what strain of rats was used, and what post-natal day were they? Play peaks at 30 – 40 days post-natal so anything older then 80 days can lead to antagonistic encounters

Line 147 –If the animals are reaching adulthood their interactions could become antagonistic, so I wonder if the social partner could become aggressive was this tested?

Line 154 – I wonder if separation calls would be the same as anticipatory calls? Also, these calls are from a male adult Wistar (again are the rats also Wistar?) and the tested animals are juveniles, I wonder if an age matched stimuli would work better? Male rats do not interact well between ages, you actually cannot test social interaction between juvenile and adult males as the adults will become so aggressive with the juveniles they can actually kill them. What effect does the stimuli call coming from an adult has on the juvenile rats? I think a small note about this would be important

Results

Was the social interaction scored? Again, to look for antagonistic interactions or to correlate the social experience with the performance in the task could be very enlightening, there is no way to know if all the interactions are the same. (this isn’t necessary but could be helpful)

Discussion

Line 381 – again please clarify the hypothesis

Line 431 – a point could be added that the social condition could be antagonistic due to the age of the animals (depending on the age)

I think a discussion point needs to be made about the potential that the stimulus is not related to gaining a social partner, but in fact one being taken away, could this cause the effect seen? Or could the fact that there is an age discrepancy in the playback stimulus and this could effect the results.

Reviewer 2 Report

Review was uploaded. 

Reviewer 3 Report

The manuscript by Berz et al. is an interesting behavioral-immunohistochemical analysis. Three groups were analyzed "Empty cage", "Social Partner", and "Group Cage".

The authors showed individual differences in social information processing and correlated behavior with c-Fos protein expression. Interestingly, despite the previous speculation that the nucleus accumbens is responsible for the social response, the activity of ACC, which is increasingly indicated to be a structure associated with social interactions, was found to be significant for the approach response.

The authors in the “animal housing section” describe the use of 36 animals, then we can read about 3 groups of 10 animals each - the experimental groups. Where did the 6 individuals go?  - were these individuals used for testing in the Social Partner group? Please add some information.

There is a lack of a drawing that describes the behavioral-biochemical procedure - and a depiction of the behavioral equipment used. If there is a link to the publication (2007?) or a drawing - it would make it easier for the reader to understand the issue. 

p.3 l. 101 – it should be strain-dependent.

p.3.l 130 – please add that author used Wistar in the sentence

p.3.l.105 I wouldn't make strong assumptions on voltammetry data where only dopamine was measured.

p.3.l149 Shouldn't the c-fos analysis be done 90 minutes after the test?

Section 2.7. Why was an ANOVA with repeated measures used? wouldn't one-way ANOVA have been better here?

p.9.l.311 bracket missing after “(Fig.7”

p.11.l.346 Why was an ANOVA with repeated measures used? wouldn't one-way ANOVA have been better here?

I would also suggest using another statistic to compare groups with each other. In addition, I would not use a bar on the figures, but boxplots and individual results in the form of dots, placed in boxplots.

Discussion part:

Despite the fact that the effect was achieved with groups "empty cage" or " group cage "  - I would put more emphasis in the discussion on social novelty - in the group "social partner" - where the rat was not a known individual and the test animal entered his territory (unfriendly environment). Was ultrasonic vocalization measured during this interaction?

p.14 l. 465 did the author mean “sensu stricto”?

Round 2

Reviewer 2 Report

Dear Authors,

Thank you for the detailed answers to my questions and suggestions.

Minor comment, due to the changes, please take care of the numbering of the figures in the main text, e.g. line 388. (Fig 9b) is incorrect. 

In the light of the additional data, it is reasonable to accept the authors' argument that, based on the behavioral data presented, the extinction or  decreased interest to the speaker during retesting cannot be explained by the absence or presence of social behavior after initial USV.

However Fig. 10 shows that the decrease in the change score during the retest is due to the decrease in the locomotor activity during the playback of the 50-kHz USV. Did you analyse statistically the potentially significant decrease before and during the playback of the 50-kHz USV? 

Such a decrease, indicative of vocal or spatial habituation to the situation, could generally explain the decrease in interest during retest.

Regarding the immunohistochemical data, I accept the Authors' answer regarding the NAcc, where a decrease in approach to the 50-kHz USV is associated with a decrease in activation of that brain area.

I also agree that post-hoc adding any experimental group to an experiment is irresponsible, they should be done together. 

However, as for the surprising immunohistochemical results in the ACC, the Authors also admit that the correlated results are from a single observation and are not comparable to a control group that was not exposed to repeated 50-kHz USV. For such observations, if there is a suitable experimental background (unfortunately not here), it is worth designing a focused experiment to avoid being so speculative in the conclusion. The increase in c-Fos number in the ACC after 50-kHz USV as described in the Sadananda et al. paper was also measured in n=4 animals, but there are no data to prove whether there is a correlation between repeated USV playback and ACC activation, as described by the authors in line 581 of the revised manuscript.

I maintain my previous opinion, without substantiated data it is only speculation, as the authors conclude in line 612 of the revised Discussion.

Author Response

Dear Reviewer,

thank you again for your thoughtful comments to which we have responded as follows:

Minor comment, due to the changes, please take care of the numbering
of the figures in the main text, e.g. line 388. (Fig 9b) is incorrect.

corrected

In the light of the additional data, it is reasonable to accept the
authors' argument that, based on the behavioral data presented, the
extinction or  decreased interest to the speaker during retesting
cannot be explained by the absence or presence of social behavior
after initial USV.

However Fig. 10 shows that the decrease in the change score during the
retest is due to the decrease in the locomotor activity during the
playback of the 50-kHz USV. Did you analyse statistically the
potentially significant decrease before and during the playback of the
50-kHz USV?

Such a decrease, indicative of vocal or spatial habituation to the
situation, could generally explain the decrease in interest during
retest.

In order to address this point, we have calculated linear regressions of locomotor activity for the 5min before versus the 5min during the retest as an index of the degree of habituation. Descriptively, the mean negative slope before playback (-7.35, indicating habituation in locomotor activity) was less pronounced than that during the subsequent 5min of playback (-15.95) but statistical comparisons between the two did not yield significant results (p=.232, 2-tailed t-test for repeated measures). Therefore, we have no significant evidence for a stronger habituation during playback as compared to immediately before. Due to the lacking significance, we would like to abstain from addressing this point in the MS.

Regarding the immunohistochemical data, I accept the Authors' answer
regarding the NAcc, where a decrease in approach to the 50-kHz USV is
associated with a decrease in activation of that brain area.

I also agree that post-hoc adding any experimental group to an
experiment is irresponsible, they should be done together.

However, as for the surprising immunohistochemical results in the ACC,
the Authors also admit that the correlated results are from a single
observation and are not comparable to a control group that was not
exposed to repeated 50-kHz USV. For such observations, if there is a
suitable experimental background (unfortunately not here), it is worth
designing a focused experiment to avoid being so speculative in the
conclusion. The increase in c-Fos number in the ACC after 50-kHz USV
as described in the Sadananda et al. paper was also measured in n=4
animals, but there are no data to prove whether there is a correlation
between repeated USV playback and ACC activation, as described by the
authors in line 581 of the revised manuscript.

I maintain my previous opinion, without substantiated data it is only
speculation, as the authors conclude in line 612 of the revised
Discussion.

We have added a paragraph at the end of the discussion to address the limitation in case of the ACC and to emphasize again that this finding is exploratory and would require further control experiments.

Finally, and as asked for by the editors we have adjusted the style of literature references to that required by the journal and we hope that the MS is now acceptable for publication.
